# Human Embryonic Stem-Cell-Derived Exosomes Repress NLRP3 Inflammasome to Alleviate Pyroptosis in Nucleus Pulposus Cells by Transmitting miR-302c

**DOI:** 10.3390/ijms24087664

**Published:** 2023-04-21

**Authors:** Yawen Yu, Wenting Li, Tinghui Xian, Mei Tu, Hao Wu, Jiaqing Zhang

**Affiliations:** 1Department of Biochemistry and Molecular Biology, School of Preclinical Medicine, Jinan University, 601 West Huangpu Avenue, Guangzhou 510632, China; 2Department of Materials Science and Engineering, Jinan University, Guangzhou 510632, China; 3Department of Orthopedic Surgery, The First Affiliated Hospital, Jinan University, Guangzhou 510632, China

**Keywords:** intervertebral disc degeneration, NLRP3 inflammasome, embryonic stem cells, exosomes, miR-302c

## Abstract

Recent studies have shown that the NOD-, LRR-, and pyrin domain-containing protein 3 (NLRP3) inflammasome is extensively activated in the process of intervertebral disc degeneration (IVDD), leading to the pyroptosis of nucleus pulposus cells (NPCs) and the exacerbation of the pathological development of the intervertebral disc (IVD). Exosomes derived from *human* embryonic stem cells (hESCs-exo) have shown great therapeutic potential in degenerative diseases. We hypothesized that hESCs-exo could alleviate IVDD by downregulating NLRP3. We measured the NLRP3 protein levels in different grades of IVDD and the effect of hESCs-exo on the H_2_O_2_-induced pyroptosis of NPCs. Our results indicate that the expression of NLRP3 was upregulated with the increase in IVD degeneration. hESCs-exo were able to reduce the H_2_O_2_-mediated pyroptosis of NPCs by downregulating the expression levels of NLRP3 inflammasome-related genes. Bioinformatics software predicted that miR-302c, an embryonic stem-cell-specific RNA, can inhibit *NLRP3*, thereby alleviating the pyroptosis of NPCs, and this was further verified by the overexpression of miR-302c in NPCs. In vivo experiments confirmed the above results in a *rat* caudal IVDD model. Our study demonstrates that hESCs-exo could inhibit excessive NPC pyroptosis by downregulating the NLRP3 inflammasome during IVDD, and miR-302c may play a key role in this process.

## 1. Introduction

Intervertebral disc degeneration (IVDD) is a chronic degenerative disease that is often associated with neck or back pain, causing significant social, economic, and clinical impacts worldwide [1,2]. The microenvironment of the intervertebral disc (IVD) is characterized by hypoxia, cyclic tension, hypertonicity, the accumulation of profibrotic and inflammatory factors, acidic conditions, and nutrient deficiency [3,4]. Recent studies showed that inflammation is gradually activated during the progress of IVDD, contributing to the deterioration of microenvironmental homeostasis [5,6]. The deterioration of the microenvironment further exacerbates the pathological changes in IVD, resulting in a decreased cell number, an abnormal cell phenotype, and the degradation of the extracellular matrix (ECM) [7]. However, the method by which to suppress the excessive inflammatory response in *human* IVDD remains unclear. Therefore, modulating inflammation may be a therapeutic target for IVDD.

Recently, accumulating evidence has indicated that the nucleotide-binding oligomerization domain (NOD)-, leucine-rich repeat (LRR)-, and pyrin domain (PYD)-containing protein 3 (NLRP3) inflammasome plays a critical role in the pathogenesis of IVDD [8,9]. Recent studies have shown that the expressions of NLRP3 inflammasome-related proteins (including NLRP3 and its downstream targets caspase-1 and interleukin-1β (IL-1β)) are upregulated in IVDD and can lead to inflammatory responses, cell pyroptosis, and the degradation of ECM components [8]. The NLRP3 inflammasome is a multiprotein complex that induces cell membrane perforation, cell rupture, and the release of contents by activating caspase-1, which then processes the proinflammatory cytokines IL-1β and IL-18 and the cleavage of gasdermin D (GSDMD), resulting in inflammatory reactions and pyroptosis [10,11].

Several experiments have demonstrated that treatment with H_2_O_2_ or LPS can in-crease the level of reactive oxygen species (ROS) in *human* nucleus pulposus cells (NPCs), leading to the expressions of NLRP3, IL-1β, and IL-18 and the subsequent pyroptosis of NPCs [12,13]. Exosomes [10], drugs (melatonin [14], honokiol [15], and MCC950 [16]), and siRNA [17] have been shown to delay the progression of IVDD by inhibiting the activation of the NLRP3 inflammasome in animal models. However, the inhibition of abnormal NLRP3 inflammasome activation in *human* IVDD requires further investigation.

*Human* embryonic stem cells (hESCs) possess the unlimited self-renewal ability and multi-differentiation potential, making them a promising therapy for degenerative diseases [18,19]. However, they have some biological and ethical restrictions [20,21,22,23]. Exosomes derived from hESCs (hESCs-exo) carry hESC-specific miRNAs, mRNAs, and proteins, offering the regenerative potential of hESCs while avoiding their immune and ethical problems [24]. Therefore, hESCs-exo have emerged as critical mediators in providing an alternate cell-free therapeutic modality [25,26,27]. Several studies have shown that exosomes from embryonic stem cells can reduce inflammation and pyroptosis, such as increasing the expressions of the anti-inflammatory cytokines IL-10, IL-4, IL-9, and IL-13 [28,29] and reducing pyroptosis in doxorubicin-induced cardiomyopathy [29]. We hypothesized that the delivery of hESCs-exo could inhibit excessive NPC pyroptosis by downregulating the NLRP3 inflammasome during IVDD.

Based on a bioinformatics analysis and further mechanism studies, our work suggests that hESCs-exo may alleviate the pyroptosis of NPCs by downregulating the NLRP3 inflammasome via the delivery of miR-302c.

## 2. Results

### 2.1. NLRP3 Inflammasome Was Excessively Activated in Nucleus Pulposus (NP) Tissues with the Grades of IVDD

We collected NP tissue samples from 20 cases (Table 1) and measured the expression of NLRP3 in each specimen. Hematoxylin and eosin (HE) staining revealed that the NPCs tended to concentrate as numerous spherical units surrounded by an abundant ECM (Figure 1A). The proteoglycan (blue) and collagen (red) concentrations in the NP tissues were assessed histologically using Alcian blue and Picrosirius red [30,31]. As shown in Figure 1A, with the grade of IVDD, the collagen staining area (red) of the NP tissues increased, while the proteoglycan area (blue) decreased. According to immunohistochemistry (IHC) and Western blot (WB) assays, the expression of NLRP3 was proportional to the grade of IVDD (II: 1.583 ± 1.504, III: 3.833 ± 1.407, and IV: 8.5 ± 2.505, Figure 1A–C). These findings suggest that the activation of the NLRP3 inflammasome occurred during IVDD.

### 2.2. hESCs-Exo Attenuate Pyroptosis of NPCs Treated with H_2_O_2_

As shown in Figure 2A, hESCs were identified by detecting the expressions of pluripotency factors SOX2 and OCT4. Western blot confirmed that hESCs-exo expressed exosomal markers CD63 and TSG101, but not the negative marker transmembrane endoplasmic reticulum chaperone calnexin (Figure 2B). Based on a cell-counting kit-8 (CCK-8) assay (Figure 2C), a 500 μM dose of H_2_O_2_ was chosen to treat the NPCs for 24 h to initiate inflammation and pyroptosis [10,15]. Obvious cell swelling, large bubbles, cell membrane rupture, mitochondrial damage, and increased autophagosomes were observed with transmission electron microscopy (TEM) under H_2_O_2_ treatment, and they were alleviated after the replacement of H_2_O_2_ with hESCs-exo (10 μg/mL) for 48 h (Figure 2D). Furthermore, the elevated mRNA expression levels of *NLRP3*, *caspase-1*, *IL-1β*, and *GSDMD* induced by the H_2_O_2_ intervention were considerably decreased in the hESCs-exo-treated group compared with the H_2_O_2_-treated group (Figure 2E). A WB analysis confirmed the above results at protein levels (Figure 2F,G). Taken together, these results indicate that hESCs-exo could alleviate the pyroptosis of NPCs by inhibiting NLRP3 inflammasome priming and activation in vitro.

### 2.3. hESCs-Exo-Derived miR-302c Targets NLRP3 to Alleviate Pyroptosis of NPCs

A Venn diagram shows the prediction results of the NLRP3 target microRNAs (miRNAs) in DIANA, miRcode, and TargetScan (Figure 3A). We performed a TargetScan software analysis and identified one potential binding site for miR-302c in the NLRP3 3′ untranslated region (UTR) at 296bp–303bp (Figure 3B). Many studies have shown that the miR-302 family is abundant in hESCs-exo [32,33]. According to Figure 3C, the miR-302c level was lower in the NPCs but increased following hESCs-exo treatment. To further study the effect of miR-302c on alleviating the pyroptosis of NPCs, a synthetic miR-302c mimic (50 nM) was transfected into the NPCs, and the transfection effect was verified using miRNA-specific quantitative real-time polymerase chain reaction (qRT-PCR) (Figure 3C). Meanwhile, MCC950, a potent, selective, and common small-molecule inhibitor of NLRP3, was used to treat the NPCs as a positive control. A CCK-8 assay showed that the increase in cell viability was amplified over time in the NPCs treated with hESCs-exo, miR-302c mimics, or MCC950, especially at 72 h (Figure 3D). The mRNA and protein expression levels of NLRP3, caspase 1, IL-1β, and GSDMD in the NPCs were significantly decreased after hESCs-exo treatment or the overexpression of miR-302c (Figure 3E,F). These results suggest that hESCs-exo could suppress NLRP3 inflammasome activation in NPCs by delivering miR-302c.

### 2.4. hESCs-Exo-Derived miR302c Could Retard IVDD in a Rat Model

To further explore the therapeutic effect of miR-302c in the process of IVDD, a *rat* model of IVDD was created by puncturing caudal IVDs with a 21 G needle. Histological evaluations were performed before surgery and at 2, 4, and 6 weeks after surgery (Figure 4A). HE staining showed that the NP tissues in the PBS (phosphate-buffered saline) group (2 μL) and the hESCs-exo + miR-302c antagomir group (100 μg/mL hESCs-exo + 5 nmol miR-302c antagomir, 2 μL) were atrophied and irregular in shape. In contrast, an injection of hESCs-exo (100 μg/mL, 2 μL) or miR-302c agomir (5 nmol, 2 μL) significantly alleviated the morphological deterioration of the NP tissues. Alcian blue and Picrosirius red staining showed that the disc ECM in the PBS and hESCs-exo + miR-302c antagomir groups was unsurprisingly degraded, while the glycosaminoglycan content of the nucleus pulposus was richer in the control, hESCs-exo, and miR-302c agomir groups (Figure 4B). IHC staining showed that the positive area of NLRP3 in the hESCs-exo group (1.3 ± 0.9154) or the miR-302c agomir group (4.567 ± 1.695) was smaller than that in the PBS group (9.3 ± 2.366) and the hESCs-exo+ miR-302c antagomir group (8.033 ± 2.251). Thus, these data suggest that hESCs-exo-derived miR-302c could reduce the NLRP3 inflammasome and ameliorate damage in a needle puncture rat model of IVDD.

## 3. Discussion

Intervertebral disc degeneration (IVDD) is a multifactorial disease that is often attributed to aging [34,35], inflammation [36,37], and other environmental stresses [38,39]. Recent studies have shown that the NLRP3 inflammasome is widely activated in the process of IVDD, and it can mediate the production of various inflammatory factors, such as IL-1β and IL-18, and further promote the process of IVDD [8,40]. The assembly of NLRP3 inflammasomes leads to the activation of caspase-1, which in turn cleaves the pro-inflammatory cytokines IL-1β and IL-18, as well as the pyroptotic substrate GSDMD [11,41]. As GSDMD cleaves, the N-terminus GSDMD (NT-GSDMD) oligomerizes, creating plasma membrane pores that promote cell death and the release of the mature forms of IL-1β and IL-18 [42].

While the surgical treatment of IVDD has certain limitations, new biological therapies, such as biomaterial-based tissue engineering, stem cell therapy, injections of growth factor, and gene therapy, show great promise [43,44]. Recent research has demonstrated that *human* embryonic stem cells (hESCs) can alleviate IVDD [21,22,45], and hESCs-exo may play a critical role in this process [24,28,46]. However, the mechanisms underlying this effect remain unclear. Therefore, in this study, we aimed to investigate whether hESCs-exo could slow down the inflammatory response in IVDD, possibly by inhibiting the abnormal activation of the inflammasome. Our findings may shed light on the potential therapeutic application of hESCs-exo in treating IVDD.

Our results demonstrate that hESCs-exo can alleviate pyroptosis in nucleus pulposus cells (NPCs) by inhibiting the activation of the NLRP3 inflammasome, thereby delaying IVDD. Compared to the H_2_O_2_ group, the H_2_O_2_ + hESCs-exo group exhibited a significant improvement in the state of NPCs, the downregulation of NLRP3 inflammasome-related genes (*NLRP3*, *caspase 1*, *GSDMD* and *IL-1β*), and an improvement in cell activity.

MiRNAs are abundant in hESCs-exo and can modulate the function of recipient cells via the post-transcriptional regulation of gene expression. Numerous studies have identified the miR-302 family as being enriched in hESCs-exo [47,48,49]. The miR-302/367 cluster consists of miR-302a, miR-302b, miR-302c, miR-302d, and miR-367, which are co-transcribed from a single polycistronic unit [50]. We hypothesized that the miR-302/367 cluster might be the main substance that inhibits the NLRP3 inflammasome in hESCs-exo. Our findings revealed that miR-302c may target the 3’UTR of NLRP3, leading to the downregulation of NLRP3 mRNA, thereby suppressing the assembly of the NLRP3 inflammasome and its biological functions in inflammation. Our results suggest that NLRP3, caspase-1, IL-1β, and GSDMD were significantly reduced in the H_2_O_2_-induced pyroptosis of NPCs following treatment with hESCs-exo or miR-302c mimics, indicating that hESCs-exo treatment might reduce pyroptosis caused by NLRP3 inflammasome activation by delivering miR-302c into NPCs.

We investigated the role of hESCs-exo-derived miR-302c in a *rat* model of IVDD. According to previous reports [51,52,53], we punctured the caudal IVDs of *rats* with a 21 G needle to create an IVDD model. To avoid accelerating the IVDD process, we used a 34 G needle to deliver PBS, hESCs-exo, miR-302c agomir, and hESCs-exo + miR-302c antagomir to the IVDD model [51,54]. The treatment with hESCs-exo + miR-302c antagomir significantly increased the expression of NLRP3 and damaged the NP tissue compared with the hESCs-exo groups. Our results show that the hESCs-exo-derived miR-302c may play a crucial role in alleviating the pyroptosis of NPCs and relieve the progression of IVDD by targeting NLRP3 signaling in vivo.

However, there are several limitations in this study. Firstly, although miR-302c alleviated the H_2_O_2_-induced pyroptosis of NPCs in our studies, the long-term therapeutic effect and possible side effects remain unclear. Secondly, further research on larger samples is required, as the data from this study are based on a small number of animals. Thirdly, although our results indicate that miR-302c can maintain NP homeostasis by inhibiting NLRP3 inflammasome activation, we need to validate our preliminary experiments in the clinic by collecting clinical specimens.

In conclusion, our work suggests that hESCs-exo-derived miR-302c may alleviate pyroptosis in NPCs to retard IVDD by suppressing NLRP3 inflammasome activation. MiR-302c might be useful for treating IVDD, which has the potential to inhibit the NLRP3 inflammasome and alleviate pyroptosis, and it is expected to address the limitations of the surgical treatment of IVDD.

## 4. Materials and Methods

### 4.1. Nucleus Pulposus Cells (NPCs) Isolation and Culture

NPCs were extracted from the NP tissues of 20 patients with IVDD (detailed information about the specimens is in Table 1). Briefly, the tissues were washed with phosphate-buffered saline (PBS, Keygen, Nanjing, China), mechanically chopped into fine pieces (1 mm^3^), and digested with 0.2% collagenase II (Sigma, St. Louis, MA, USA) for 2 h at 37 °C [54,55]. Afterward, tissue debris was removed by passing through a 70 µm strainer. The filtrate was centrifuged at 1000 rpm for 5 min to collect the cells. After washing with PBS, the cellular pellets were cultured in DEME/F12 (Gibco, Billings, MT, USA) supplemented with 10% fetal bovine serum (FBS, PAN, Aidenbach, Germany) and 1% penicillin/streptomycin (Keygen, Nanjing, China). When the density rises between about 80 and 90%, NPCs can be passed with trypsin (Keygen, Nanjing, China). The NPCs used in the experiments were at passage 3.

### 4.2. Human Embryonic Stem Cells (hESCs) Culture

*Human* embryonic stem cell line H9 was provided by the National key Institute of Stem Cells, School of Medicine, Sun Yat-sen University, China. The hESCs were maintained on a Matrigel (Cellapy, Beijing, China) coated 25 cm^2^ flask (Nest, Wuxi, China) with a PSCeasy^®^II medium (Cellapy, China) at 37 °C in 5% CO_2_.

### 4.3. Human Embryonic Stem Cell-Derived Exosomes (hESCs-Exo) Isolation, Purification, and Characterization

Every 24 h, the hESCs culture medium (hESCs-CM) was collected for hESCs-exo extraction. The isolation of hESCs-exo was performed following the initial centrifugation steps of hESCs-CM at 300× *g*, 2000× *g* at 4 °C to remove cellular debris followed via filtration through a 0.22 µm sterilized filter membrane (Millipore, Burlington, MA, USA) [56]. The resulting supernatant was subjected to ultracentrifugation (Beckman Coulter, Brea, CA, USA) for 1 h at 100,000× *g* at 4 °C followed by a second ultracentrifugation for 1 h at 100,000× *g* at 4 °C after washing the pellet with PBS [56]. The hESCs-exo were then measured using a BCA protein assay (Beyotime, Shanghai, China) after being resuspended in PBS and filtered through a 0.22 µm sterilized filter membrane. Western blot determined the expressions of the exosome markers, CD63 (1:1000, Millipore, Burlington, MA, USA) and TSG-101 (1:1000, Termo Fisher, Waltham, MA, USA). The entire culture medium alone served as the control group, whereas the NPCs cultivated with the complete culture medium, including 10 μg/mL hESCs-exo, for 48 h were referred to as the hESCs-exo group.

### 4.4. hESCs-Exo Treatment of NPCs Induced by H_2_O_2_ Pyroptosis

The NPCs were divided into three groups: the control group, the H_2_O_2_ group, and the H_2_O_2_+hESCs-exo group. In addition to the control group, the other two groups of cells were treated with an appropriate concentration of H_2_O_2_ for 24 h, and then the medium containing H_2_O_2_ was replaced. The H_2_O_2_+hESCs-exo group was cultured with the medium containing 10 μg/mL hESCs-exo for 48 h, and the other two groups were cultured with an ordinary medium for 48 h at the same time. After 48 h, the three groups of NPCs were used for subsequent experiments.

### 4.5. Cell Transfection

The miR-302c mimic and negative controls (NC) were from RiboBio, Guangzhou, China. The miR-302c mimic (50 nM) and NCs (50 nM) were transfected into the NPCs using a transfection reagent according to the manufacturer’s instructions for 48 h.

### 4.6. Transmission Electron Microscopy (TEM)

After the culture medium was removed, the NPCs were collected into centrifuge tubes, prefixed with a TEM fixative (Servicebio, Wuhan, China) for 2 h, embedded using 1% agarose, and then postfixed with 1% osmium tetroxide (Ted Pella Inc., Redding, CA, USA) for 2 h. Following dehydration with graded ethanol, penetration with an acetone resin (Sinaopharm Group Chemical Reagent Co. Ltd., Shanghai, China), and embedding with EMBed 812 (SPI, East Petersburg, PA, USA), the NPCs were sliced into thin slices using an ultramicrotome (Leica, UC7, Wetzlar, Germany). We observed the sections with Hitachi HT7800 transmission electron microscopy (Hitachi, Tokyo, Japan) at 120 kV after staining them with 2% uranyl acetate and 6% lead citrate.

### 4.7. Quantitative Real-Time Polymerase Chain Reaction (qRT-PCR)

The total RNA was extracted from the NPCs using RNAiso (Takara, Kyoto, Japan). According to the manufacturer’s protocol, reverse-transcribed (RT) complementary DNA (cDNA) was synthesized with the Evo M-MLV RT Premix (Accurate Biology, Changsha, China). For qRT-PCR, a CFX96 Touch™ Real-Time PCR Detection System (BIO-RAD, Hercules, CA, USA) was used in conjunction with 2x SYBR Green qPCR Master Mix (Selleck, Houston, TX, USA). To quantify miR-302c expression, an RT reaction and qPCR were performed using a miDETECTA Track™ Uni-RT Primer and a miDETECTA Track™ miRNA qRT-PCR Starter kit (RiboBio, Guangzhou, China). GAPDH and U6 served as controls for mRNA and miRNA levels, respectively. The 2^−ΔΔCT^ method was used to calculate the relative expression levels. The primer sequences are listed in Table 2.

### 4.8. Western Blot (WB)

*Human* NP tissues or cells were homogenized using a RIPA buffer (Beyotime, Shanghai, China) supplemented with a protease and phosphatase inhibitor cocktail (Solarbio, Beijing, China). After clarification via centrifugation, the protein concentration was measured using a BCA protein assay (Beyotime, China) as described in the manufacturer’s instructions. The protein extracts (20 μg) were separated via electrophoresis on 8~12% sodium dodecyl sulfate-polyacrylamide gels (SDS-PAGE) and transferred to polyvinylidene fluoride membranes (PVDF, Millipore, Burlington, MA, USA). The membranes were then blocked for 1.5 h with 5% (*w*/*v*) non-fat milk after being washed with Tris-buffered saline with Tween 20 (TBST, 0.05% Tween 20). After that, a variety of primary antibodies were applied to the membranes overnight at 4 °C. The membranes were then washed with TBST and treated for 1 h at 37 °C with the matching HRP-conjugated secondary antibodies from Cell Signaling Technology (CST, Fall River, MA, USA). Finally, the protein bands were detected using an Immobilon Western Chemiluminescent HRP Substrate (Millipore, Burlington, MA, USA) as per the manufacturer’s specifications and analyzed using Image J 1.46r software. The relative protein expression was obtained via GAPDH for normalization. The primary antibodies against these molecules were used as follows: GAPDH (1:5000, Proteintech, 10494-1-AP, Rosemont, IL, USA), NLRP3 (1:1000, Affinity Biosciences, DF7438, Changzhou, China), cleaved-caspase-1 (Asp296) (1:1000, Affinity Biosciences, AF4005, China), cleaved-IL-1β (Asp116) (1:1000, Affinity Biosciences, AF4006, China), and NT-GSDMD (1:1000, Affinity Biosciences, AF4012, China).

### 4.9. Cell-Counting Kit-8 (CCK-8) Assay

Cell proliferation was detected using a cell-counting kit-8 assay (CCK-8; Beyotime, China) following the manufacturer’s instructions. NPCs from various treatment groups were planted at a density of 9000 cells per well into 96-well plates. Next, 100 μL of DMEM/F12 media was mixed with 10 μL of a CCK8 solution, and the mixture was then incubated at 37 °C for 3 h. A cell growth curve was produced using the fold changes in absorbance recorded at 450 nm in comparison to the control group.

### 4.10. Immunofluorescence (IF) Analysis

The cells were fixed in 4% paraformaldehyde for 15 min at room temperature. The cells were washed three times with PBS before being permeabilized for 15 min with 0.5% Triton X-100 and then blocked for 30 min with goat serum. The cells were incubated with primary antibodies against SOX2 (1:200, CST, #3579, USA) and OCT4 (1:200, CST, #2750, USA) overnight at 4 °C. Subsequently, the cells were incubated for 1h at 37 °C with a secondary antibody conjugated to Alexa Fluor^®^488 or Alexa Fluor^®^555 (CST, 1:200, USA). After washing, the nuclei were stained for 5 min in the dark with 4,6-diamidino-2-phenylindole (DAPI, Beyotime, China). The fluorescence was visualized under a confocal microscope (Leica, Wetzlar, Germany, DMi8, 72 dpi).

### 4.11. Rat IVDD Model

The Laboratory Animal Ethics Committee of Jinan University authorized the animal experimentation methods. Twelve three-month-old Sprague Dawley (SD) rats from Charles River, China, served as the model and appropriate controls for disc degeneration. The SD rats were anesthetized intraperitoneally with 2% (*w*/*v*) pentobarbital (40 mg/kg), and the rat intervertebral disc was located by slitting the skin surface of the rat tail. The intervertebral disc was punctured with a 21 G needle at a depth of 5 mm, rotated 360°, and remained inside the disc for 30 S before being removed. More specifically, Co3/4 was used as a control, and Co4/5, Co6/7, Co8/9, and Co10/11 were used as IVDs, which were treated with an injection of PBS, hESCs-exo (100 μg/mL), hESCs-exo + miR-302c antagomir (5 nmol), and miR-302c agomir (5 nmol), respectively, using a 34 G needle (Hamilton, Bonaduz, Switzerland) with the same total injection volume (2 μL), in the 2nd week after needle puncture. Histological evaluations were performed before surgery and at 2, 4, and 6 weeks after surgery (3 rats each).

### 4.12. Histological Valuation

Human NP tissues and rat IVDs were processed for paraffin embedding and sectioning into 5 μm sections for histological staining or immunohistochemistry (IHC). The cellular morphology was examined using hematoxylin and eosin (HE) staining. Alcian blue and Picrosirius red with hematoxylin counter staining was further performed on the sections to examine the structural, cellularity, and ECM changes in the human NP tissues and the rat IVDs. A team of skilled histology specialists used a microscope to blindly evaluate the cellularity and morphology of the nucleus pulposus and annulus fibrosus and then to make an assessment using a grading scheme, as previously mentioned [57,58].

For IHC, the sections were deparaffinized, rehydrated, and then heated for 30 min in sodium citrate before being allowed to cool for 30 min. Then, 3% hydrogen peroxide in methanol was used to suppress endogenous peroxidase activity for 20 min. Sections were incubated with normal goat serum for 1 h at room temperature to block the binding of the non-specific protein. The primary antibody (NLRP3, 1:200, Adipogen, AG-20B-0014, San Diego, CA, USA) was incubated with the sections for an entire night at 4 °C. At 37 °C for 1 h, and the secondary antibody goat against rabbit/mouse IgG-HRP (CST, 1:200, USA) was added. Then, the sections were stained with DAB (ORiGENE, Wuxi, China) for 1 min and hematoxylin for 10s. After washing with PBS, the sections were examined under a fluorescence microscope (Leica, Wetzlar, Germany, DM6000B, 300 dpi). The IHC results were further analyzed using a semiquantitative method [58,59]. Each disc was observed in 5 fields of view using Leica Application Suite V4 (LAS AF, Wetzlar, Germany) software. The number of stained positive cells and the staining intensity were used to produce scores, and the two scores (range: 0–12) were multiplied to obtain the protein expression intensity. All sections were analyzed by two independent observers blinded to the experimental details.

### 4.13. Statistical Analysis

Data are expressed as mean ± SD. Differences between groups were evaluated using Student’s *t*-test or a one-way analysis of variance (ANOVA), followed by Tukey’s test. *p* < 0.05 was considered statistically significant.

## Figures and Tables

**Figure 1 ijms-24-07664-f001:**
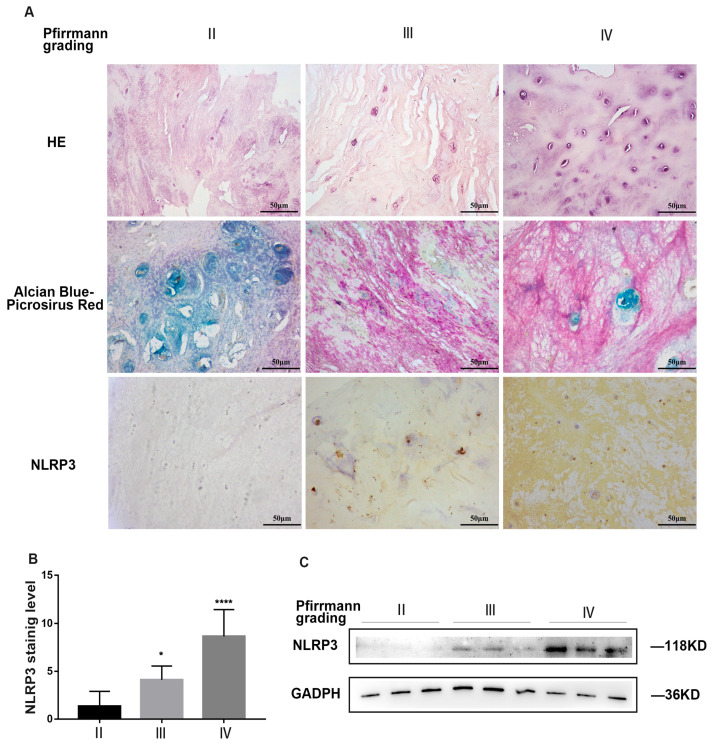
NLRP3 inflammasome was excessively activated in NP tissues with the grades of IVDD (**A**). HE, Alcian blue/Picrosirius red staining, and IHC staining of NLRP3 in different degenerative NP tissues (II–IV groups according to Pfirrmann grade), n = 3. (**B**). Statistical analysis of NLRP3 immunohistochemistry of NP tissues with different degeneration grades. (**C**). Western blot of NLRP3 protein in different degenerative NP tissues. The data are shown as the means  ± SD, n = 3. *, compared to II group; * *p* < 0.05, **** *p* < 0.0001.

**Figure 2 ijms-24-07664-f002:**
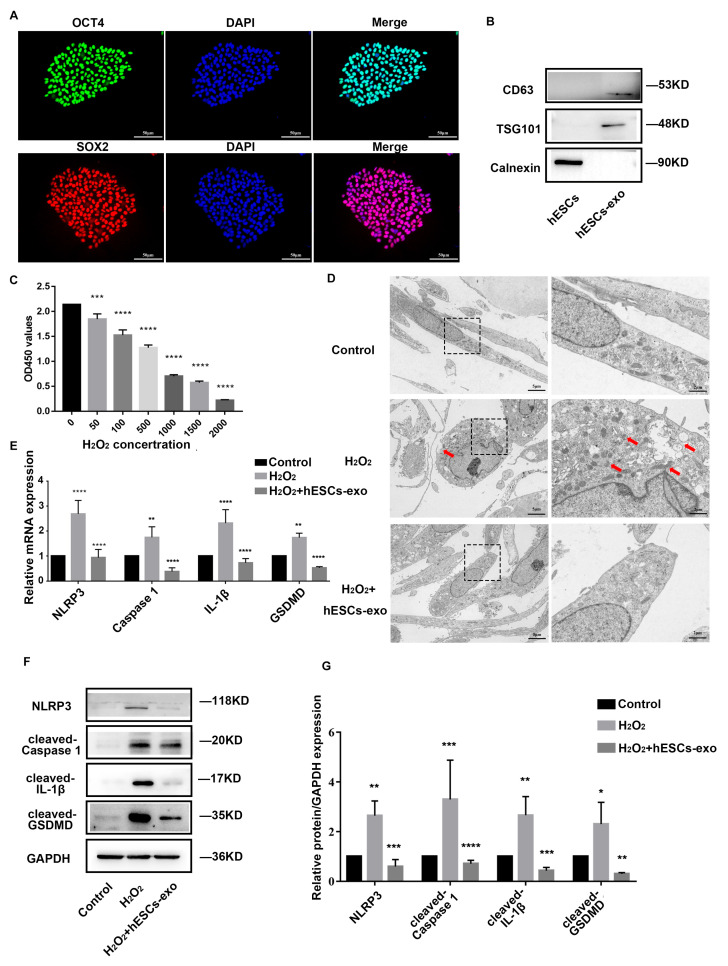
hESCs-exo attenuate pyroptosis of NPCs treated with H_2_O_2_. (**A**). IF staining analysis of pluripotency-related markers SOX2 and OCT4 of hESCs, n = 3. (**B**). WB showed the presence of exosomes markers, including CD63 and TSG101, but not the negative marker calnexin, n = 3. (**C**). Viability of NPCs cultured with 0–2000 μM H_2_O_2_ for 24 h was evaluated using CCK-8. The data are shown as the means ± SD, n = 5. Compared to the control group; *** *p* < 0.001, **** *p* < 0.0001. (**D**). Representative TEM images of NPCs of control and treatment groups of H_2_O_2_ and H_2_O_2_ + hESCs-exo. The red arrow shows the characterization of pyroptosis of NPCs. Scale bar: 5 μm and 2 μm, n = 3. (**E**). Quantitative real-time polymerase chain reaction (qRT-PCR) was used to evaluate the expressions of *NLRP3*, *caspase 1*, *IL-1β*, and *GSDMD* in NPCs after corresponding treatments. The data are shown as the means ± SD, n = 3. (**F**,**G**). WB analysis of NLRP3, cleaved-caspase 1, cleaved-IL-1β, and cleaved-GSDMD in NPCs after corresponding treatments. The data are shown as the means ± SD, n = 3. The H_2_O_2_ group compared to the control group, the H_2_O_2_ + hESCs-exo group compared to the H_2_O_2_ group; * *p* < 0.05, ** *p* < 0.01, *** *p* < 0.001. **** *p* < 0.0001.

**Figure 3 ijms-24-07664-f003:**
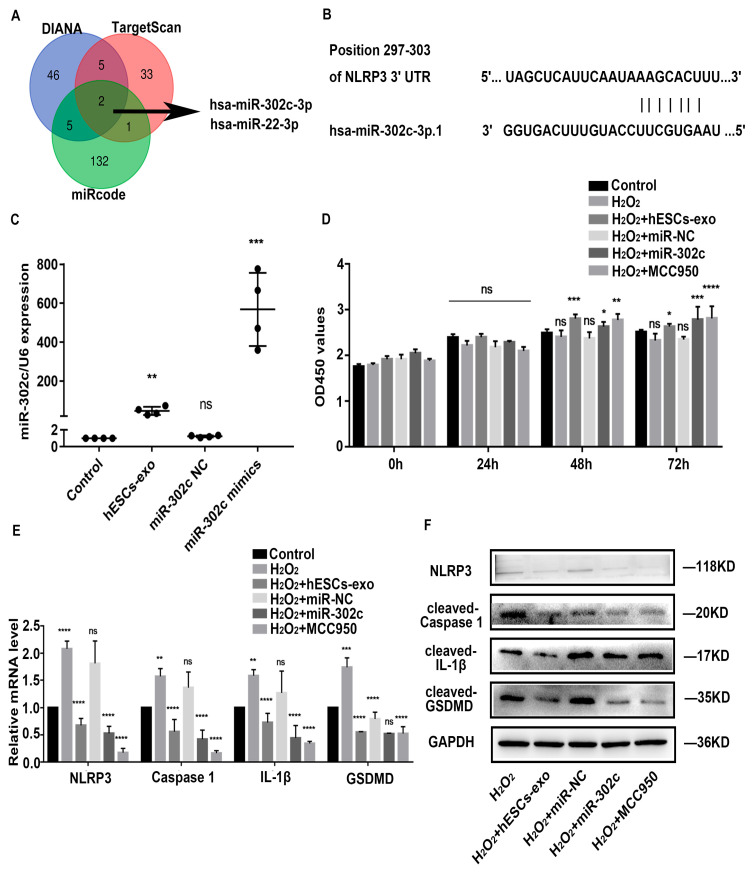
hESCs-exo-derived miR-302c targets NLRP3 to alleviate pyroptosis of NPCs. (**A**). Venn graph showing the prediction results of NLRP3 targets in DIANA, miRcode, and TargetScan software packages. (**B**). Sequence alignment of one putative miR-302c-binding site within the 3′UTR of NLRP3 mRNA shows a high level of sequence conservation and complementarity with miR-302c. (**C**). qRT-PCR was used to evaluate the expression of miR-302c in NPCs treated with corresponding treatments, n = 4. The data are shown as the means ± SD. ns, non-significant; the hESCs-exo group compared to the control group, the miR-302c negative controls (NC) group compared to the control group, the miR-302c mimic group compared to the miR-302c NC group. (**D**). CCK8 assay was used to evaluate the cell viability in NPCs treated with corresponding treatments, n = 5. (**E**). qRT-PCR was used to evaluate the expressions of *NLRP3*, *caspase 1*, *IL-1β*, and *GSDMD* in NPCs after corresponding treatments, n = 3. (**F**). WB analysis of NLRP3, cleaved-caspase 1, cleaved-IL-1β, and cleaved-GSDMD in NPCs after corresponding treatments, n = 3. The data are shown as the means ± SD. The H_2_O_2_ group compared to the control group, the H_2_O_2_ + hESCs-exo group compared to the H_2_O_2_ group, the H_2_O_2_ + miR-302c NC group compared to the H_2_O_2_ group, the H_2_O_2_ + miR-302c mimic group compared to the H_2_O_2_ + miR-302c NC group, the H_2_O_2_ + MCC950 group compared to the H_2_O_2_ group; ns, non-significant, * *p* < 0.05, ** *p* < 0.01, *** *p* < 0.001, **** *p* < 0.0001.

**Figure 4 ijms-24-07664-f004:**
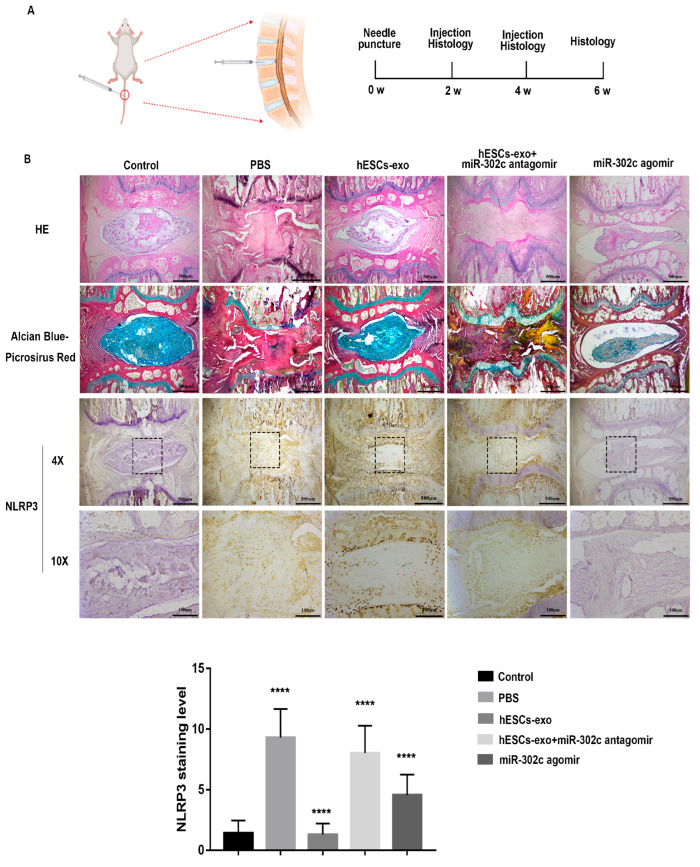
hESCs-exo-derived miR302c could retard IVDD in a rat model. (**A**). Image of the construction of a disc degeneration model created with BioRender.com. (**B**). Images of HE, Alcian blue/Picrosirius red staining, IHC staining of NLRP3, and qualification analysis in the different groups are shown. The data are shown as the means ± SD, n = 3. The PBS group compared to the control group, the hESCs-exo group compared to the PBS group, the hESCs-exo + miR-302c antagomir group compared to the hESCs-exo group, the miR-302c agomir group compared to the PBS group. **** *p* < 0.0001.

**Table 1 ijms-24-07664-t001:** Summary of patients’ information.

Case No.	Age (Years)	Gender	Diagnosis	Disc Level	Pfirrmann Grading
1	45	M	Lumbar disc herniation	L4/5	II
2	52	F	Lumbar disc herniation	L5/S1	IV
3	53	M	Lumbar disc herniation	L4/5	IV
4	31	M	Lumbar disc herniation	L4/5	II
5	66	F	Lumbar disc herniation	L4/5	III
6	42	F	Lumbar disc herniation	L4/5	IV
7	47	F	Lumbar disc herniation	L4/5	III
8	24	M	Lumbar disc herniation	L5/S1	II
9	43	M	Lumbar disc herniation	L4/5	III
10	49	M	Lumbar disc herniation	L4/5	IV
11	25	M	Lumbar disc herniation	L3/4	II
12	40	F	Lumbar disc herniation	L4/5	IV
13	59	F	Lumbar disc herniation	L3/4	III
14	30	M	Lumbar disc herniation	L4/5	III
15	67	F	Lumbar disc herniation	L4/5	IV
16	52	M	Lumbar disc herniation	L4/5	III
17	54	F	Lumbar disc herniation	L4/5	III
18	72	M	Lumbar disc herniation	L3/4	III
19	69	F	Lumbar disc herniation	L4/5	IV
20	26	M	Lumbar disc herniation	L5/S1	II

**Table 2 ijms-24-07664-t002:** The primer sequence of target genes.

Target Gene	Forward	Reverse
*GAPDH*	GGGAGCCAAAAGGGTCAT	GAGTCCTTCCACGATACCAA
*NLRP3*	CAACCTCACGTCACACTGCT	TTTCAGACAACCCCAGGTTC
*Caspase 1*	TTTCCGCAAGGTTCGATTTTCA	GGCATCTGCGCTCTACCATC
*GSDMD*	GACCCTAACACCTGGCAGAC	CACCTCAGTCACCACGTACAC
*IL-1β*	TTCGACACATGGGATAACGAGG	TTTTTGCTGTGAGTCCCGGAG
hsa-miR-302c mimics	UAAGUGCUUCCAUGUUUCAGUGG	AUUCACGAAGGUACAAAGUCACC
hsa-miR-302c mimics NC	UUUGUACUACACAAAAGUACUG	AAACAUGAUGUGUUUUCAUGAC

## Data Availability

The data presented in this study are available upon reasonable request from the corresponding author.

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
