# Peer review of "Human Embryonic Stem-Cell-Derived Exosomes Repress NLRP3 Inflammasome to Alleviate Pyroptosis in Nucleus Pulposus Cells by Transmitting miR-302c"

_ijms, 2023, doi:10.3390/ijms24087664_

Round 1

Reviewer 1 Report

The abstract should be rewritten. I can not find sufficient, reasonable, and strong reasons why the authors would like to explore the role exosome derived from the human ESC plays in the inflammatory process in the IVDD. Besides, the reasons eliciting IVDD have many controversial interpretations. Not all scenarios elicit the inflammatory disease process. Furthermore, I can not find the abstract's most important concluded results and novel scientific findings. Therefore, the motivation, hypothesis, unmet needs, and rationale to initiate this study are unclear. This should also be delineated in the introduction. 

The NLRP3 expression is the consequence of IVDD or the initiator of IVDD. Therefore, the authors must interpret, convey, and persuade the readers to believe.

In the section "2.1. NLRP3 inflammasome was excessively activated in IVD with the grades of IVDD", what was the disc degeneration model? IRB number? Identical to the scenario of the puncture disc model? The sample number and individual experiment number should be stated in Figure 1.

In the section "2.2. hESCs-exo attenuates pyroptosis of NPCs treated with H2O2", the NPC derived from should be stated. In addition, how many quantities of exo were used should be noted. And the sample number and individual experiment number should be reported.

"2.3. hESCs-exo derived miR-302c targets NLRP3 to alleviate pyroptosis of NPCs". What data evidence does the exo contain for the mir-302c? The sample number and individual experiment number should be stated.

2.4. hESCs-exo-derived miR302c could retard IVDD in a Rat model. How did the author demonstrate the NLRP3 overexpression? From the staining level, density, or histomorphological analysis, please the readers to believe and show the raw data to the reviewers. How many quantities of exo administrated in the coccygeal disc should be stated. The sample number and individual experiment number should be noted.

In the discussion, please delineate the novel findings for the study hypothesis, scientific discoveries, and achievements, and solve unmet medical needs.

Author Response

Dear reviewer:

Thank you for your constructive comments on my manuscript. As you are concerned, there are several problems that need to be addressed. We have tried our best to improve and made some changes in the revised manuscript. The revised manuscript has been carefully revised by a native English speaker to improve the grammar and readability. In this revised version, changes to our manuscript were all highlighted within the document by using red-colored text. Revision notes, point-to-point, are given as follows:

Point 1: The abstract should be rewritten. I can not find sufficient, reasonable, and strong reasons why the authors would like to explore the role exosome derived from the human ESC plays in the inflammatory process in the IVDD. Besides, the reasons eliciting IVDD have many controversial interpretations. Not all scenarios elicit the inflammatory disease process. Furthermore, I can not find the abstract's most important concluded results and novel scientific findings. Therefore, the motivation, hypothesis, unmet needs, and rationale to initiate this study are unclear. This should also be delineated in the introduction.

Response 1: Thanks for your suggestion. We have rewritten the Abstract and Introduction part according to your suggestion as follows:

Abstract:

Recent studies have shown that NLRP3 (NOD-, LRR- and pyrin domain-containing protein 3) in-flammasome is extensively activated in the process of intervertebral disc degeneration (IVDD), leading to pyroptosis of nucleus pulposus cells (NPCs) and exacerbates the pathological develop-ment of intervertebral disc (IVD). Exosomes derived from human embryonic stem cells (hESCs-exo) have shown great therapeutic potential in degenerative diseases. We hypothesized that hESCs-exo could alleviate IVDD by downregulating NLRP3. We measured the NLRP3 protein level in different grades of IVDD and the effect of hESCs-exo on H2O2-induced pyroptosis of NPCs. Our results in-dicated that the expression of NLRP3 was up-regulated with the increase of IVD degeneration. hESCs-exo were able to reduce H2O2-mediated pyroptosis of NPCs by downregulating the expression level of NLRP3 inflammasome-related genes. Bioinformatics software predicted that miR-302c, an embryonic stem cell-specific RNA, and overexpression of miR-302c further verified that it can inhibit NLRP3, thereby alleviating pyroptosis of NPCs. In vivo experiments confirmed the above results in a rat caudal IVDD model. Our study demonstrated that hESCs-exo could inhibit excessive NPCs pyroptosis by downregulating the NLRP3 inflammasome during IVDD, and miR-302c may play a key role in this process.

Introduction:

Intervertebral disc degeneration (IVDD) is a chronic degenerative disease that is often associated with neck or back pain, causing significant social, economic, and clinical impact worldwide [1, 2]. The microenvironment of the intervertebral disc (IVD) is characterized by hypoxia, cyclic tension, hypertonicity, accumulation of profibrotic and inflammatory factors, acidic conditions, and nutrient deficiency [3, 4]. Recent studies showed that inflammation is gradually activated during the progress of IVDD, contributing to the deterioration of microenvironmental homeostasis [5, 6]. The deterioration of the microenvironment further exacerbates the pathological changes of IVD, resulting in decreased cell number, abnormal cell phenotype, and degradation of the extracellular matrix (ECM)[7]. However, how to suppress the excessive inflammatory response in human IVDD remains unclear. Therefore, modulating inflammation may be a therapeutic target for IVDD.

Recently, accumulating evidence has indicated that the nucleotide-binding oligomerization domain (NOD)-, leucine-rich repeat (LRR)- and pyrin domain (PYD)- containing protein 3 (NLRP3) inflammasome plays a critical role in the pathogenesis of IVDD[8, 9]. Recent studies have shown that the expression of NLRP3 inflammasome-related proteins (including NLRP3 and its downstream targets caspase-1 and IL-1β) is upregulated in IVDD and can lead to inflammatory responses, cell pyroptosis, and degradation of ECM components[8]. The NLRP3 inflammasome is a multiprotein complex that induces cell membrane perforation, cell rupture, and release of contents by activating caspase-1, which then processes the proinflammatory cytokines interleu-kin-1β (IL-1β), IL-18 and cleavage of gasdermin D (GSDMD), resulting to inflammatory reactions and pyroptosis[10, 11].

Several experiments have demonstrated that treatment with H2O2 or LPS can in-crease the level of reactive oxygen species (ROS) in human nucleus pulposus cells (NPCs), leading to the expression of NLRP3, IL-1β, and IL-18 and subsequent pyroptosis of NPCs[12, 13]. Exosomes[10], drugs (melatonin[14], honokiol[15], MCC950[16]), and siRNA[17] have been shown to delay the progression of IVDD by inhibiting the activation of the NLRP3 inflammasome in animal models. However, the inhibition of abnormal NLRP3 inflammasome activation in human IVDD requires further investigation.

Human embryonic stem cells (hESCs) possess the unlimited self-renewal ability and multi-differentiation potential, making them a promising therapy for degenerative diseases[18, 19]. However, they have some biological and ethical restrictions[20-23]. Exosomes derived from hESCs carry hESCs-specific miRNAs, mRNAs, and proteins, offering the regenerative potential of hESCs while avoiding their immune and ethical problems[24]. Therefore, hESCs-exo have emerged as critical mediators in providing an alternate cell-free therapeutic modality[25-27]. Several studies have shown that exosomes from embryonic stem cells can reduce inflammation and pyroptosis, such as increasing the expression of the anti-inflammatory cytokine IL-10, IL-4, IL-9, and IL-13[28, 29], and reducing pyroptosis in doxorubicin-induced cardiomyopathy[29]. We hypothesized that the delivery of hESCs-exo could inhibit excessive NPCs pyroptosis by downregulating the NLRP3 inflammasome during IVDD.

Based on bioinformatics analysis and further mechanism studies, our work suggests that hESCs-exo may alleviate pyroptosis of NPCs by downregulating the NLRP3 inflammasome through delivering miR-302c.

Point 2: The NLRP3 expression is the consequence of IVDD or the initiator of IVDD. Therefore, the authors must interpret, convey, and persuade the readers to believe.

Response 2: Thanks for your suggestion. We have illustrated the relationship between NLRP3 and IVDD in the Introduction part carefully as described as following at Line 45-61 Page2:

Recently, accumulating evidence has indicated that the nucleotide-binding oligomerization domain (NOD)-, leucine-rich repeat (LRR)- and pyrin domain (PYD)- containing protein 3 (NLRP3) inflammasome plays a critical role in the pathogenesis of IVDD [8, 9]. Many studies have shown that the expression of NLRP3 inflammasome-related proteins (including NLRP3 and its downstream targets caspase-1 and IL-1β) is upregulated in IVDD and can lead to inflammatory responses, cell pyroptosis, and degradation of ECM components [8]. The NLRP3 inflammasome is a multiprotein complex that induces cell membrane perforation, cell rupture, and release of contents by activating caspase-1, which then processes the proinflammatory cytokines interleu-kin-1β (IL-1β), IL-18, and cleavage of gasdermin D (GSDMD), resulting to inflammatory reactions and pyroptosis [10, 11].

Several experiments have demonstrated that treatment with H2O2 or LPS can in-crease the level of reactive oxygen species (ROS) in human nucleus pulposus cells (NPCs), leading to the expression of NLRP3, IL-1β, and IL-18 and subsequent pyroptosis of NPCs [12, 13]. Exosomes [10], drugs (melatonin [14], honokiol [15], MCC950[16]), and siRNA [17] have been shown to delay the progression of IVDD by inhibiting the activation of the NLRP3 inflammasome in animal models. However, the inhibition of abnormal NLRP3 inflammasome activation in human IVDD requires further investigation.

Moreover, we have added more references on NLRP3 and IVDD into the Introduction part of the revised manuscript.

Point 3: In the section "2.1. NLRP3 inflammasome was excessively activated in IVD with the grades of IVDD", what was the disc degeneration model? IRB number? Identical to the scenario of the puncture disc model? The sample number and individual experiment number should be stated in Figure 1.

Response 3: Thank you for pointing it out. In the section 2.1. NLRP3 inflammasome was determined in the NP tissue samples collected from patients who underwent micro endoscopic discectomy for IVDD, and assessed the degree of IVDD by the pfirrmann grading system. The Pfirrmann grading system is the most widely known classification for intervertebral disc degeneration and is used in both clinical and research fields (detailed information in Table 1/ Figure 1) [1]. The study was conducted in accordance with the Declaration of Helsinki, and the protocol was approved by the IRB of the First Affiliated Hospital of Jinan University (KY-2023-060). The IRB number has been showed in the Institutional Review Board Statement section of the article (Page12 Line 397-401). In vivo experiments of this study, we used acupuncture to create disc degeneration in rats. Acupuncture is one of the most commonly used methods to establish animal models of IVD degeneration, and the depth of puncturing the intervertebral disc is controlled according to the size of the needle and the height of the intervertebral disc [2,3]. Although not completely reproducing the clinical situation, animal studies may provide useful information for human subjects at earlier stages of the degenerative process [4]. Recent studies have shown that a smaller needle (21 G) is sufficient to cause significant disc degeneration, but there is no difference in qualitative analysis based on mechanics, biochemical content, disc height, and histology [5-6]. Therefore, the fibrous annulus puncture method was used in this experiment to simulate the process of intervertebral disc degeneration such as protrusion, dehydration and degeneration of the nucleus pulposus after annulus fibrosus injury. In order to minimize the impact of the operation itself on the intervertebral disc, a 21 G needle puncture depth of 5 mm and a 34 G needle injection were used in our study. The animal study protocol was approved by the Laboratory Animal Welfare and Ethics Committee of Jinan University (protocol code: IACUC-20210303-06, 03/03/2021 approval).

The sample number and individual experiment number were added in Figure 2.1 legend.

The following are relevant references:

  1. Pfirrmann CW, Metzdorf A, Zanetti M, Hodler J, Boos N. Magnetic resonance classification of lumbar intervertebral disc degeneration. Spine (Phila Pa 1976). 2001 Sep 1;26(17):1873-8.
  2. Zhang H, Yang S, Wang L, Park P, La Marca F, Hollister SJ, Lin CY. Time course investigation of intervertebral disc degeneration produced by needle-stab injury of the rat caudal spine: laboratory investigation. J Neurosurg Spine. 2011 Oct;15(4):404-13.
  3. Keorochana G, Johnson JS, Taghavi CE, Liao JC, Lee KB, Yoo JH, Ngo SS, Wang JC. The effect of needle size inducing degeneration in the rat caudal disc: evaluation using radiograph, magnetic resonance imaging, histology, and immunohistochemistry. Spine J. 2010 Nov;10(11):1014-23.
  4. Cunha C, Lamas S, Gonçalves RM, Barbosa MA. Joint analysis of IVD herniation and degeneration by rat caudal needle puncture model. J Orthop Res. 2017 Feb;35(2):258-268. doi: 10.1002/jor.23114. Epub 2016 Nov 23. PMID: 26610284.
  5. Chen F, Jiang G, Liu H, Li Z, Pei Y, Wang H, Pan H, Cui H, Long J, Wang J, Zheng Z. Melatonin alleviates intervertebral disc degeneration by disrupting the IL-1β/NF-κB-NLRP3 inflammasome positive feedback loop. Bone Res. 2020 Feb 18;8:10.
  6. Zhang J, Li Z, Chen F, Liu H, Wang H, Li X, Liu X, Wang J, Zheng Z. TGF-β1 suppresses CCL3/4 expression through the ERK signaling pathway and inhibits intervertebral disc degeneration and inflammation-related pain in a rat model. Exp Mol Med. 2017 Sep 22;49(9):e379.

Fig 1 A–E, The Grading system for the assessment of lumbar disc degeneration. Grade I: The structure of the disc is homogeneous, with a bright hyperintense white signal intensity and a normal disc height. Grade II: The structure of the disc is inhomogeneous, with a hyperintense white signal. The distinction between the nucleus and annulus is clear, and the disc height is normal, with or without horizontal gray bands. Grade III: The structure of the disc is inhomogeneous, with an intermediate gray signal intensity. The distinction between nucleus and anulus is unclear, and the disc height is normal or slightly decreased. Grade IV: The structure of the disc is inhomogeneous, with a hypointense dark gray signal intensity. The distinction between the nucleus and annulus is lost, and the disc height is normal or moderately decreased. Grade V: The structure of the disc is inhomogeneous, with a hypointense black signal intensity. The distinction between the nucleus and annulus is lost, and the disc space is collapsed. Grading is performed on T2-weighted midsagittal (repetition time 5000 msec/echo time 130 msec) fast spin-echo images.

Point 4: In the section "2.2. hESCs-exo attenuates pyroptosis of NPCs treated with H2O2", the NPC derived from should be stated. In addition, how many quantities of exo were used should be noted. And the sample number and individual experiment number should be reported.

Response 4: Thanks for your suggestion. NPCs derived from the NP tissues of 20 IVDD patients who had micro endoscopic discectomy, the detailed information was described on Line 243-252, Page 8 in the Materials and Methods Section. And quantities of hESCs-exo were added on Line 105, Page 3. The sample number and individual experiment number were added in Figure 2.2 legend.

Point 5: "2.3. hESCs-exo derived miR-302c targets NLRP3 to alleviate pyroptosis of NPCs". What data evidence does the exo contain for the mir-302c? The sample number and individual experiment number should be stated.

Response 5: Thanks for your suggestion. Previous studies have shown that the miR-302 family is abundant in hESCs-exo. We have checked the literature carefully and added the related references to the Discussion part in the revised manuscript.

Meanwhile, we also detected the level of miR-302c in the control (without treatment) group and hESCs-exo treatment group in NPCs by miRNA-specific qRT-PCR, and found that the level of miR-302c was less in NPCs, but increased after hESCs-exo treatment (Fig 3C). The sample number and individual experiment number were added in Figure 2.3 legend.

Reference:

  1. Suh MR, Lee Y, Kim JY, Kim SK, Moon SH, Lee JY, Cha KY, Chung HM, Yoon HS, Moon SY, Kim VN, Kim KS. Human embryonic stem cells express a unique set of microRNAs. Dev Biol. 2004 Jun 15;270(2):488-98.
  2. Ren J, Jin P, Wang E, Marincola FM, Stroncek DF. MicroRNA and gene expression patterns in the differentiation of human embryonic stem cells. J Transl Med. 2009 Mar 23;7:20.

Point 6: 2.4. hESCs-exo-derived miR302c could retard IVDD in a Rat model. How did the author demonstrate the NLRP3 overexpression? From the staining level, density, or histomorphological analysis, please the readers to believe and show the raw data to the reviewers. How many quantities of exo administrated in the coccygeal disc should be stated. The sample number and individual experiment number should be noted.

Response 6:

We sincerely thank you for pointing it out. Immunohistochemical staining showed that compared with the hESCs-exo group (1.3 ± 0.9154) and miR-302c agomir group (4.567 ± 1.695), the positive staining area of NLRP3 was significantly increased in both the PBS group (9.3 ± 2.366) and the hESCs-exo+miR-302c antagomir group (8.033 ± 2.251). These results indicated that miR302c could reduce the expression of NLRP3 and retard IVDD in a Rat model. The above has been added to this section of Results 2.4 (Line 169-171, Page 6). We have sent the raw data of immunohistochemistry to the assistant editor. Regarding the dosage of drugs in animal experiments, we have added in the Method part (Line 353-357, Page 11) and also supplemented relevant content in the Result part (Line 161-175, Page 6). The sample number and individual experiment number were added in Figure 2.4 legend.

Point 7: In the discussion, please delineate the novel findings for the study hypothesis, scientific discoveries, and achievements, and solve unmet medical needs.

Response 7: Thanks for your suggestion. We have rewritten the Discussion part of the revised manuscript according to your suggestion.

Here is our rewritten discussion:

Intervertebral disc degeneration (IVDD) is a multifactorial disease that is often attributed to aging[34, 35], inflammation[36, 37], and other environmental stresses[38, 39]. Recent studies have shown that NLRP3 inflammasome is widely activated in the process of IVDD, which can mediate the production of various inflammatory factors, such as IL-1β and IL-18, and further promote the process of IVDD[8, 40]. NLRP3 inflammasome assembly results in caspase-1 activation, which in turn leads to caspase-1 activation, which in turn cleaves the proinflammatory cytokines IL-1β and IL-18, as well as the pyroptotic substrate GSDMD [11, 41]. As GSDMD cleaves, the N-terminus (N-GSDMD) oligomerizes, creating plasma membrane pores that promote cell death and release of the mature forms of IL-1β and IL-18[42].

While surgical treatment of IVDD has certain limitations, new biological therapies such as biomaterial-based tissue engineering, stem cell therapy, injections of growth factor, and gene therapy show great promise[43, 44]. Recent research has demonstrated that human embryonic stem cells (hESCs) can alleviate IVDD[21, 22, 45], and hESCs-exo may play a critical role in this process[24, 28, 46]. However, the mechanisms underlying this effect remain unclear. Therefore, in this study, we aimed to investigate whether hESCs-exo could slow down the inflammatory response in IVDD, possibly by inhibiting the abnormal activation of the inflammasome. Our findings may shed light on the potential therapeutic application of hESCs-exo in treating IVDD.

Our results demonstrated that hESCs-exo can alleviate pyroptosis in nucleus pulposus cells (NPCs) by inhibiting the activation of the NLRP3 inflammasome, thereby delaying IVDD. Compared to the H2O2 group, the H2O2 + hESCs-exo group exhibited significant improvement in the state of NPCs, downregulation of NLRP3 inflammasome-related genes (NLRP3, Caspase 1, GSDMD, IL-1β, and IL-18), and improvement in cell activity.

MiRNAs are abundant in hESCs-exo and can modulate the function of recipient cells by post-transcriptional regulation of gene expression. Numerous studies have identified the miR-302 family as being enriched in hESCs-exo[47-49]. The miR-302/367 cluster consists of miR-302a, miR-302b, miR-302c, miR-302d and miR-367, which are co-transcribed from a single polycistronic unit[50]. We hypothesized that the miR-302/367 cluster might be the main substance that inhibits the NLRP3 inflammasome in hESCs-exo. Our findings revealed that miR-302c tar-gets the 3'UTR of NLRP3, leading to the downregulation of NLRP3 mRNA, thereby suppressing the assembly of the NLRP3 inflammasome and its biological functions in inflammation. Our results suggested that NLRP3, Caspase-1, IL-1β, and GSDMD were significantly reduced in H2O2-induced pyroptosis of NPCs following treatment with hESCs-exo or miR-302c mimics, indicating that hESCs-exo treatment might reduce pyroptosis caused by NLRP3 inflammasome activation by delivering miR-302c into NPCs.

We investigated the role of hESCs-exo-derived miR-302c in a rat model of IVDD. According to previous reports[51-53], we punctured the caudal IVDs of rats with a 21 G needle to create an IVDD model. To avoid accelerating the IVDD process, we used a 34 G needle to deliver PBS, hESCs-exo, miR-302c agomir, and hESCs-exo + miR-302c antagomir to the IVDD model [51, 54]. Treatment with hESCs-exo + miR-302c antagomir significantly increased the expression of NLRP3 and damaged the NP tissue com-pared with the hESCs-exo groups. Our results showed that the hESCs-exo derived miR-302c may play a crucial role in alleviating pyroptosis of NPCs and relieve the progression of IVDD through targeting NLRP3 signaling in vivo.

However, there are several limitations in this study. Firstly, although miR-302c alleviates H2O2-induced pyroptosis of NPCs in our studies, the long-term therapeutic effect and possible side effects remain unclear. Secondly, further research on larger samples is required, as the data from this study are based on a small number of animals. Thirdly, although our results indicated that miR-302c can maintain NP homeostasis by inhibiting NLRP3 inflammasome activation, we need to validate our preliminary experiments in the clinic by collecting clinical specimens.

In conclusion, our work suggests that hESCs-exo derived miR-302c alleviates pyroptosis in NPCs to retard IVDD by suppressing NLRP3 inflammasome activation. MiR-302c might be useful for treating IVDD, which has the potential to inhibit NLRP3 inflammasome and alleviate pyroptosis and is expected to address the limitations of surgical treatment of IVDD.

We hope that the revised manuscript is suitable for publication in International Journal of Molecular Sciences. Thank you again for your help with the manuscript.

Thank you and best regards.

Yours sincerely,

Ms. Yawen Yu

E-Mail: yyw929@stu2020.jnu.edu.cn

Reviewer 2 Report

Thank you for this very nice report. You claim that IVD degeneration in humans has increasing amounts of inflammasome activation and subsequently pyroptosis, and that miR-302c delivery, based on hESC-exosome content analysis, decreases inflammasome activity. Furthermore, a rat IVD puncture model is rescued by miR-302C delivery.

I find that this study has broad merits. The data are simply and effectively laid out, and the writing is succint and logical. A few questions

1. Human disc herniation fragments are likely degenerated, but are also known to be highly inflammatory, and as such, may correlate with a rat injury model. They may not be representative of human IVD degeneration (without herniation).  I'm not aware of literature describing the differences in inflammasome activation between disc herniation fragments and IVD degeneration tissue. Can you please comment? This is important for any translational work that may arise; IVD degeneration samples would have to be tested for inflammasome activity

2. Please reference human NP cell extraction and culture techniques. 

3. Please reference hESC exosome extraction methodology. This is important.

4. Rat model: -can you clarify numbers? You had 12 animals who served as internal controls. Presumably, you had 3 animals at each time-point?

-where is the histological scoring data?

-please describe the LAS AF software suite-generated NLRP3 staining intensity level measurement. Reference if there is a previous description.

-while the histology and NLRP3 levels are reasonable estimates of mi-R302c modification of the injury model, why was there no real-time PCR or WB blot performed on IVD to demonstrate inflammasome / pyroptosis activity?  I can understand that you'd at least need to double the sample size for such data. Or why no cell death estimates?

Author Response

Dear reviewer:

Thank you for your constructive comments on my manuscript. As you are concerned, there are several problems that need to be addressed. We have tried our best to improve and made some changes in the revised manuscript. The revised manuscript has been carefully revised by a native English speaker to improve the grammar and readability. In this revised version, changes to our manuscript were all highlighted within the document by using red-colored text. Revision notes, point-to-point, are given as follows:

Point 1: Human disc herniation fragments are likely degenerated, but are also known to be highly inflammatory, and as such, may correlate with a rat injury model. They may not be representative of human IVD degeneration (without herniation). I'm not aware of literature describing the differences in inflammasome activation between disc herniation fragments and IVD degeneration tissue. Can you please comment? This is important for any translational work that may arise; IVD degeneration samples would have to be tested for inflammasome activity.

Response 1: Thank you for your positive comments. We have carefully revised the whole manuscript according to your comments.

Many studies have found that IVDD is closely related to pyroptosis[1]. Pyroptosis is triggered primarily by a variety of inflammasomes, and is executed by caspases and gasdermin proteins [2]. The NLRP3 inflammasome in the NLR family is the most studied in relation to IVDD [3-4]. De Rivero Vaccari JP and his colleagues obtained healthy nucleus pulposus cells from the intervertebral discs of male volunteers who had never experienced spinal degenerative diseases. They found that under acidic conditions (pH = 6.5), the activation of inflammasomes in nucleus pulposus cells was reduced. Other stimuli besides increased acidification, like altered oxygenation or altered glucose levels, may also be the triggers for the increased inflammatory responses that are found in the NP during IVDD [5]. Cao Yang et al. found that the expression of NLRP3, Caspase 1, and IL-1β was significantly increased in degenerative disc tissue samples compared with normal tissue (obtained from patients with idiopathic scoliosis or vertebral fracture) [6]. Recent studies have shown that the expression of NLRP3 inflammasome-related proteins (including NLRP3 and its downstream targets caspase-1 and IL-1β) was upregulated with increased of IVD degeneration, and can lead to inflammatory responses, cell pyroptosis, and degradation of ECM components [7-8]. In general, abnormal activation of inflammasome was less in non-herniated intervertebral disc degeneration, and the expression of inflammasome-related genes gradually increased with the degree of intervertebral disc degeneration. IVD degeneration samples might need to be tested for inflammasome activity.

Due to factors, such as animal ethics, source, costs, human similarity, and ease of manipulation, the most commonly used animals for establishing IVDD models are rats/mice and rabbits. The most typical mechanical damage model is the rat caudal IVD acupuncture model, which is currently also the most widely applied model [9-10]. Although not completely reproducing the clinical situation, animal studies may provide useful information for human subjects at earlier stages of the degenerative process.

References:

  1. Zhou J, Qiu J, Song Y, Liang T, Liu S, Ren C, Song X, Cui L, Sun Y. Pyroptosis and degenerative diseases of the elderly. Cell Death Dis. 2023 Feb 9;14(2):94.
  2. Schroder K, Tschopp J. The inflammasomes [J]. Cell, 2010, 140(6): 821-32.
  3. Chao-Yang G, Peng C, Hai-Hong Z. Roles of NLRP3 inflammasome in intervertebral disc degeneration. Osteoarthritis Cartilage. 2021 Jun;29(6):793-801.
  4. Luo J, Yang Y, Wang X, Chang X, Fu S. Role of Pyroptosis in Intervertebral Disc Degeneration and Its Therapeutic Implications. Biomolecules. 2022 Dec 2;12(12):1804.
  5. Brand FJ 3rd, Forouzandeh M, Kaur H, Travascio F, de Rivero Vaccari JP. Acidification changes affect the inflammasome in human nucleus pulposus cells. J Inflamm (Lond). 2016 Aug 24;13(1):29.
  6. Zhao K, An R, Xiang Q, Li G, Wang K, Song Y, Liao Z, Li S, Hua W, Feng X, Wu X, Zhang Y, Das A, Yang C. Acid-sensing ion channels regulate nucleus pulposus cell inflammation and pyroptosis via the NLRP3 inflammasome in intervertebral disc degeneration. Cell Prolif. 2021 Jan;54(1): e12941.
  7. Tang P, Zhu R, Ji WP, Wang JY, Chen S, Fan SW, Hu ZJ. The NLRP3/Caspase-1/Interleukin-1β Axis Is Active in Human Lumbar Cartilaginous Endplate Degeneration. Clin Orthop Relat Res. 2016 Aug;474(8):1818-26.
  8. Chen ZH, Jin SH, Wang MY, Jin XL, Lv C, Deng YF, Wang JL. Enhanced NLRP3, caspase-1, and IL- 1β levels in degenerate human intervertebral disc and their association with the grades of disc degeneration. Anat Rec (Hoboken). 2015 Apr;298(4):720-6.
  9. Cunha C, Lamas S, Gonçalves RM, Barbosa MA. Joint analysis of IVD herniation and degeneration by rat caudal needle puncture model. J Orthop Res. 2017 Feb;35(2):258-268.
  10. Su Q, Cai Q, Li Y, Ge H, Zhang Y, Zhang Y, Tan J, Li J, Cheng B, Zhang Y. A novel rat model of vertebral inflammation-induced intervertebral disc degeneration mediated by activating cGAS/STING molecular pathway. J Cell Mol Med. 2021 Oct;25(20):9567-9585.

Point 2: Please reference human NP cell extraction and culture techniques.

Response 2: Thanks for your suggestion. We have added more references on human NPCs extraction and culture on Line 246 Page 8 into the Materials and Methods part of the revised manuscript.

Reference:

  1. Liao Z, Luo R, Li G, Song Y, Zhan S, Zhao K, Hua W, Zhang Y, Wu X, Yang C. Exosomes from mesenchymal stem cells modulate endoplasmic reticulum stress to protect against nucleus pulposus cell death and ameliorate intervertebral disc degeneration in vivo. Theranostics. 2019 May 31;9(14):4084-4100.
  2. Yang SH, Hu MH, Wu CC, Chen CW, Sun YH, Yang KC. CD24 expression indicates healthier phenotype and less tendency of cellular senescence in human nucleus pulposus cells. Artif Cells Nanomed Biotechnol. 2019 Dec;47(1):3021-3028.

Point 3: Please reference hESC exosome extraction methodology. This is important.

Response 3: Thanks for your suggestion. We have checked the literature carefully and added more references on hESCs-exo extraction on Line 264 Page 9 into the Materials and Methods part in the revised manuscript.

Reference:

  1. Zhu Q, Ling X, Yang Y, Zhang J, Li Q, Niu X, Hu G, Chen B, Li H, Wang Y, Deng Z. Embryonic Stem Cells-Derived Exosomes Endowed with Targeting Properties as Chemotherapeutics Delivery Vehicles for Glioblastoma Therapy. Adv Sci (Weinh). 2019 Feb 1;6(6):1801899.
  2. Ke Y, Fan X, Hao R, Dong L, Xue M, Tan L, Yang C, Li X, Ren X. Human embryonic stem cell-derived extracellular vesicles alleviate retinal degeneration by upregulating Oct4 to promote retinal Müller cell retrodifferentiation via HSP90. Stem Cell Res Ther. 2021 Jan 7;12(1):21.

Point 4: Rat model: -can you clarify numbers? You had 12 animals who served as internal controls. Presumably, you had 3 animals at each time-point? where is the histological scoring data? Please describe the LAS AF software suite-generated NLRP3 staining intensity level measurement. Reference if there is a previous description. While the histology and NLRP3 levels are reasonable estimates of miR-302c modification of the injury model, why was there no real-time PCR or WB blot performed on IVD to demonstrate inflammasome / pyroptosis activity? I can understand that you'd at least need to double the sample size for such data. Or why no cell death estimates?

Response 4:

We sincerely thank you for your suggestion. The IVDD animal model used 3 rats per time point for immunohistochemical and histological evaluations, which have been supplemented in the Methods section as following: The intervertebral disc was punctured with a 21 G needle at a depth of 5 mm, rotated 360°, and remained inside the disc for 30 S before being removed. More specifically, Co3/4 was used as a control, Co4/5, Co6/7, Co8/9, and Co10/11 as IVDDs, which were treated with injection of PBS, hESCs-exo (100 μg/mL), hESCs-exo + miR-302c antagomir (5 nmol), and miR-302c agomir (5 nmol) respectively, using a 34 G needle (Hamilton, Switzerland) with the same total injection volume (2 μL), at 2nd week after needle puncture. Histological evaluations were performed before surgery and at 2, 4, and 6 weeks after surgery (3 rats each) respectively.

We have revised the previous erroneous description of the statistical analysis method of immunohistochemistry and cited relevant literature (Line 377-382, Page 11). The raw data of immunohistochemistry was sent to the assistant editor. We chose to measure NLRP3-specific staining and HE in tissues that most directly reflected pyroptosis to assess the overall situation of IVD degeneration. However, due to the limited number of experimental animal samples, qPCR, WB and dead cells were not used at the same time. This is also the shortcoming of this study, which will be further explored in future studies.

We hope that the revised manuscript is suitable for publication in International Journal of Molecular Sciences. Thank you again for your help with the manuscript.

Thank you and best regards.

Yours sincerely,

Ms. Yawen Yu

E-Mail: yyw929@stu2020.jnu.edu.cn

Reviewer 3 Report

Human embryonic stem cell-derived exosomes repress NLRP3 inflammasome to alleviate pyroptosis in nucleus pulposus cells by transmitting miR-302c claims exosome from human embryonic stem cell inhibits NLRP3 inflammasome and IVD degeneration. MiR-302c plays a key role to regulate the NLRP3 expression level to decrease the inflammasome and therefore delays the process of IVDD. As claimed in the paper, “after the replacement of H2O2 with hESC-exo for 48 h”, inflammation observations were relieved. This experiment is problematic. 1, the inflammation may recover when the H2O2 was removed w/o the existence of hESC-exo. And 48 hours later, it is normal to see some recovery of the inflammation. Please clearly describe this experiment in the method. 2, hESC-exo may physically decrease the H2O2 activity or through some other possibilities, but not its unlimited self-renewal ability. Extra evidence will be necessary to support the authors’ claim. Some other exosome w/o treatment effect will be ideal as a control to treat the IVDD. If it is not possible, the authors may claim that hESC-exo may facilitate the healing of the inflammation. The control will be very easy to get when removing H2O2 from the media. 3, Is miR302c expressed consistently in all the human embryonic stem cell exosomes? Does it also exist in other animal ESC exosomes or human specific? Does miR-302c target rat NLRP3 and human NLRP3? 4, More experiments are needed to support that miR302c directly regulates NLRP3 expression. 

Author Response

Dear reviewer:

Thank you for your constructive comments on my manuscript. As you are concerned, there are several problems that need to be addressed. We have tried our best to improve and made some changes in the revised manuscript. The revised manuscript has been carefully revised by a native English speaker to improve the grammar and readability. In this revised version, changes to our manuscript were all highlighted within the document by using red-colored text. Revision notes, point-to-point, are given as follows:

Point 1: the inflammation may recover when the H2O2 was removed w/o the existence of hESC-exo. And 48 hours later, it is normal to see some recovery of the inflammation. Please clearly describe this experiment in the method.

Response 1: We sincerely thank you for your careful reading. The methodology involved in issues mentioned by the reviewer has been added to the revised manuscript (Line 274-281 Page 9). We conducted a concentration gradient experiment to determine the H2O2 concentration, as shown in Figure 2C, and a 500 μM dose of H2O2 was chosen to use in the following experiment, because, at this concentration, the cell viability could not recover even after the removal of H2O2, but the addition of hESCs-Exo helped to repair cell viability.

Here is our supplementary experimental method:

4.4. hESCs-exo treatment of NPCs induced by H2O2 pyroptosis

NPCs were divided into three groups: the control group, the H2O2 group and the H2O2+hESCs-exo group. In addition to the control group, the other two groups of cells were treated with an appropriate concentration of H2O2 for 24 h, and then the medium containing H2O2 was replaced. The H2O2+hESCs-exo group was cultured with the medium containing 10 μg/mL hESCs-exo for 48 h, and the other two groups were cultured with ordinary medium for 48 hours at the same time. After 48 h, the three groups of NPCs were used for subsequent experiments.

Point 2: hESC-exo may physically decrease the H2O2 activity or through some other possibilities, but not its unlimited self-renewal ability. Extra evidence will be necessary to support the authors’ claim. Some other exosome w/o treatment effect will be ideal as a control to treat the IVDD. If it is not possible, the authors may claim that hESC-exo may facilitate the healing of the inflammation. The control will be very easy to get when removing H2O2 from the media.

Response 2: Thank you for pointing it out. As you mentioned, hESC-exo physically decrease the H2O2 activity not through its unlimited self-renewal ability but what hESC-exo contains. We set up three groups: untreated NPCs as control, the H2O2 treatment group and the H2O2+ hESC-exo group. It would certainly be better if other exosome treatment groups served as controls. However, due to limited conditions, we only set the non-exosome treatment group as control here. Therefore, we have modified the claim that hESC-exo may facilitate the healing of the inflammation as your suggestion. We have clarified this concept in the Discussion section (Line 193-206 Page 7) as follows:

While surgical treatment of IVDD has certain limitations, new biological therapies such as biomaterial-based tissue engineering, stem cell therapy, injections of growth factor, and gene therapy show great promise [43, 44]. Recent research has demonstrat-ed that human embryonic stem cells (hESCs) can alleviate IVDD [21, 22, 45], and hESCs-exo may play a critical role in this process [24, 28, 46]. However, the mechanisms underlying this effect remain unclear. Therefore, in this study, we aimed to in-vestigate whether hESCs-exo could slow down the inflammatory response in IVDD, possibly by inhibiting the abnormal activation of the inflammasome. Our findings may shed light on the potential therapeutic application of hESCs-exo in treating IVDD.

Our results demonstrated that hESCs-exo can alleviate pyroptosis in nucleus pul-posus cells (NPCs) by inhibiting the activation of the NLRP3 inflammasome, thereby delaying IVDD. Compared to the H2O2 group, the H2O2 + hESCs-exo group exhibited significant improvement in the state of NPCs, downregulation of NLRP3 inflammasome-related genes (NLRP3, Caspase 1, GSDMD, IL-1β, and IL-18), and improvement in cell activity.

Point 3: Is miR302c expressed consistently in all the human embryonic stem cell exosomes? Does it also exist in other animal ESC exosomes or human specific? Does miR-302c target rat NLRP3 and human NLRP3?

Response 3: Thank you for valuable comments. Previous studies have shown that the miR-302 family is abundant in mouse and human embryonic stem cells, especially in their exosomes. We have checked the literature carefully and added more references on miR-302c and hESCs-exo into the Result part on Line 129 Page 5 in the revised manuscript. The miR-302/367 cluster, generally consisting of five members, miR-367, miR-302d, miR-302a, miR-302c, and miR-302b, is ubiquitously distributed in vertebrates. Through bioinformatics analysis, we found that miR-302c targets the 3'UTR of human NLRP3 mRNA (Fig.3B). To date, no studies have identified miR-302c in rat embryonic stem cell exosomes, nor have miR-302c been found to target NLRP3 in rats. In the literature search, it was found that rat NLRP3 is homologous to human NLRP3, which may be the reason why mir-302c can target NLRP3 to treat IVDD in rats. In our study, miR-302c agomir had a better therapeutic effect on acupuncture-induced acute IVDD in the tail of rats, and immunohistochemistry showed that miR-302c could reduce the expression of NLRP3.

Point 4: More experiments are needed to support that miR302c directly regulates NLRP3 expression.

Response 4: Thanks for your suggestion. Our findings revealed that miR-302c targets the 3'UTR of NLRP3, leading to the downregulation of NLRP3 mRNA, thereby suppressing the assembly of the NLRP3 inflammasome and its biological functions in inflammation. We tested its effect on NLRP3 after overexpressing miR-302c in human NPCs in vitro. Data from qPCR, CCK8 and WB showed that miR-302c could reduce the expression of NLRP3 inflammasome-related genes, thereby alleviating the pyroptosis of NPCs induced by H2O2. In the animal model of IVDD, we injected PBS, hESCs-exo, miR-302c agomir, and hESCs-exo + miR-302c antagomir into the tail intervertebral disc of rats to further explore the therapeutic effect of miR-302c on IVDD in vivo. Immunohistochemistry staining showed that the positive area of NLRP3 in the hESCs-exo group or miR-302c agomir group was smaller than that the PBS group and the hESCs-exo + miR-302c antagomir group. Based on above results, we suggest that hESCs-exo-derived miR-302c may alleviate pyroptosis in NPCs to retard IVDD by suppressing NLRP3 inflammasome activation in vivo and vitro. Further, at this point, unfortunately, we are unable to continue our work due to difficult circumstances and limited resources (disruptions due to infrastructural changes and lack of research funding). In future studies, we will further explore the influence of miR-302c on NLRP3 from more aspects.

We hope that the revised manuscript is suitable for publication in International Journal of Molecular Sciences. Thank you again for your help with the manuscript.

Thank you and best regards.

Yours sincerely,

Ms. Yawen Yu

E-Mail: yyw929@stu2020.jnu.edu.cn

Round 2

Reviewer 1 Report

The authors replied to our questions and revised the manuscript accordingly; the manuscript is acceptable for publication.

Only one another comment, the figures are too small to read, especially since the scale bar is so tiny, and the label is vague and difficult to read. The figures could be more significant, and the scale bar should be more prominent. 

Author Response

Dear reviewer:

Thank you for your precious comments on our manuscript. We have tried our best to improve and made some changes in the revised manuscript. Revision notes, point-to-point, are given as follows:

Point 1: Only one another comment, the figures are too small to read, especially since the scale bar is so tiny, and the label is vague and difficult to read. The figures could be more significant, and the scale bar should be more prominent.

Response 1: Thanks for your suggestion. We have modified the sizes of all figures and the scale bar in the revised manuscript according to your suggestions.

We appreciate for Reviewers’ warm work earnestly, and hope that the revised manuscript is suitable for publication in International Journal of Molecular Sciences. Thank you very much for your kind help and good advice.

Thank you and best regards.

Yours sincerely,

Ms. Yawen Yu

E-Mail: yyw929@stu2020.jnu.edu.cn